# The *Bph45* Gene Confers Resistance against Brown Planthopper in Rice by Reducing the Production of Limonene

**DOI:** 10.3390/ijms24021798

**Published:** 2023-01-16

**Authors:** Charng-Pei Li, Dong-Hong Wu, Shou-Horng Huang, Menghsiao Meng, Hsien-Tzung Shih, Ming-Hsin Lai, Liang-Jwu Chen, Kshirod K. Jena, Sherry Lou Hechanova, Ting-Jyun Ke, Tai-Yuan Chiu, Zong-Yuan Tsai, Guo-Kai Chen, Kuan-Chieh Tsai, Wei-Ming Leu

**Affiliations:** 1Crop Science Division, Taiwan Agricultural Research Institute, Council of Agriculture, Taichung 41362, Taiwan; 2Department of Plant Protection, Chiayi Agricultural Experiment Station, Taiwan Agricultural Research Institute, Chiayi 60044, Taiwan; 3Graduate Institute of Biotechnology, National Chung Hsing University, Taichung 40227, Taiwan; 4Applied Zoology Division, Taiwan Agricultural Research Institute, Council of Agriculture, Taichung, 41362, Taiwan; 5Institute of Molecular Biology, National Chung Hsing University, Taichung 40227, Taiwan; 6Gene Identification and Validation (GIV) Laboratory, Rice Breeding Innovation Platform, International Rice Research Institute, DAPO Box 7777, Metro Manila 1301, Philippines; 7School of Biotechnology, KIIT University, Bhubaneswar 751024, Odisha, India; 8Biotechnology Center, National Chung Hsing University, Taichung 40227, Taiwan

**Keywords:** limonene, brown planthopper (BPH), antixenosis, *Oryza nivara*, Tainung 71 (TNG71), Tainan 11 (TN11), near-isogenic line (NIL), integrated pest management (IPM), terpene synthase (TPS), volatile organic compound (VOC)

## Abstract

Brown planthopper (BPH), a monophagous phloem feeder, consumes a large amount of photoassimilates in rice and causes wilting. A near-isogenic line ‘TNG71-*Bph45*’ was developed from the *Oryza sativa japonica* variety ‘Tainung 71 (TNG71) carrying a dominant BPH-resistance locus derived from *Oryza nivara* (IRGC 102165) near the centromere of chromosome 4. We compared the NIL (TNG71-*Bph45*) and the recurrent parent to explore how the *Bph45* gene confers BPH resistance. We found that TNG71-*Bph45* is less attractive to BPH at least partially because it produces less limonene. Chiral analysis revealed that the major form of limonene in both rice lines was the L-form. However, both L- and D-limonene attracted BPH when applied exogenously to TNG71-*Bph45* rice. The transcript amounts of limonene synthase were significantly higher in TNG71 than in TNG71-*Bph45* and were induced by BPH infestation only in the former. Introgression of the *Bph45* gene into another *japonica* variety, Tainan 11, also resulted in a low limonene content. Moreover, several dominantly acting BPH resistance genes introduced into the BPH-sensitive IR24 line compromised its limonene-producing ability and concurrently decreased its attractiveness to BPH. These observations suggest that reducing limonene production may be a common resistance strategy against BPH in rice.

## 1. Introduction

Rice (*Oryza sativa* L.) is extensively cultivated in diverse ecosystems ranging from tropical to temperate areas throughout the world. More than half of the world’s population relies on rice as a staple food. The brown planthopper (BPH; *Nilaparvata lugens* Stål, Hemiptera, Delphacidae) is a monophagous herbivore that feeds exclusively on rice sap and causes substantial rice yield losses in Asia and even worldwide by causing direct and indirect damage. Both nymphs and adults of BPH suck sap from the lower portion of the rice plant, which results in a reduction in chlorophyll and protein content in leaves and the rate of photosynthesis, eventually causing leaf yellowing, a reduction in tiller number, the production of unfilled grains, or complete drying and plant death, which is known as “hopper burn”. BPH also causes indirect harvest losses by transmitting rice viruses including grassy stunt and ragged stunt viruses. The cultivation of resistant rice varieties is the most economical, effective, and environmentally friendly strategy to manage BPH. The cultivation of such varieties will also help to conserve natural enemies, minimize pesticide use, and prevent stronger virulent BPH strains from evolving because of the continuous use of insecticides [1,2].

Owing to the breakdown of several BPH-resistance genes (*Bph1*, *bph2*, *Bph3*, *Bph4*, etc.), used frequently in South-East Asia due to the evolution of virulent BPH biotypes [1], the identification of new sources of BPH-resistance genes is critical for rice breeding programs. Wild species in the genus of *Oryza* are potentially valuable sources of BPH-resistance genes. Of the 1003 wild rice varieties screened by Sarao et al. [3], 159 wild rice accessions including seven *Oryza nivara* accessions were identified as promising resources for BPH resistance breeding. Although only one BPH resistance gene has been reported in *O. nivara* so far [4], *O. nivara* is actually an ideal candidate to cross with the prevalently cultivated rice species *O. sativa* because they share similar AA genomes [5], which facilitates homologous chromosome pairing and gene transfer using conventional breeding. Genetic mapping based on a segregated population derived from *O. nivara* IRGC 102165 (W33) revealed a BPH resistance gene, designated *Bph45* in this study. The *Bph45* gene locus provided a major genetic effect on BPH resistance up to 50% Phenotypic Variation Explained (PVE), and was closely linked with SSR markers RM3317 and RM16655 on chromosome 4 [6]. One of the BPH resistance introgression lines ‘852T034’ carrying the *Bph45* gene was used as the donor parent to cross with a susceptible Taiwan elite japonica variety ‘Tainung 71’ (TNG71) to establish its near-isogenic line ‘TNG71-*Bph45*’.

To date, at least 44 BPH-resistance genes have been mapped or cloned from wild species or *indica* varieties of rice, with many located on the short arm of chromosome 4, namely *Bph3*, *QBph4*, *Bph12*, *Bph15*, *Bph17*, *Bph20*(t), *Bph30*, *Bph33*, *Bph35*, and *Bph36*, as reviewed by Muduli et al. [2], and the recently published *Bph42* [7]. Five genes were reported to be on the long arm of chromosome 4: *Bph6* from the Bangladesh *indica* variety Swarnalata (~21.4 Mb) [8], *Bph27*(t) from the *indica* variety Balamawee (~21.3 Mb) [9], another *Bph27* from *Oryza rufipogon* Griff. accession no. 2183 (~19.2 Mb) [10], *Bph34* from *O. nivara* acc. IRGC104646 (~21.4 Mb) [4], and *Bph38* from *O. rufipogon* Griff. (~15 Mb) [11]. The *Bph45* gene locus reported in this study is located close by RM16655 and RM3317 (~13.7 Mb) and is distal from all gene loci reported above. Moreover, the gene source is *O. nivara*, a wild rice species from which no BPH resistance genes have ever been identified [3] except for *Bph34* [4]. These studies suggest that *Bph45* represents a novel gene conferring resistance to BPH, which makes it worthy of further investigation.

To develop integrated pest management (IPM) strategies for rice cultivation, it is crucial to understand how rice acts against BPH. The resistance mechanisms of plants are categorized as antixenosis, antibiosis, and tolerance, although multiple types of resistance could be conferred by a single BPH resistance gene [1]. Antixenosis or non-preference is a common strategy to avoid BPH infestation and was first reported in 1970 [12]. It refers to resistance mechanisms whereby the host plant has features that are unattractive or even repellent to the pests and therefore affect their settling, colonization, and/or oviposition [13,14,15,16]. Although several genes have been identified to confer antixenosis against BPH [17,18,19], their downstream gene products and how they modify plant features, hypothetically including color, texture, structure, and odor, that directly affect the behavior of BPH remain unknown.

In this report, we compared BHP resistance between the BPH-susceptible variety TNG71 and the BPH-resistance NIL TNG71-*Bph45* and found that the latter was less attractive to BPH. Among the volatile organic compounds (VOCs) emitted by rice, limonene was present at a significantly lower level in TNG71-*Bph45*. The Y-maze olfactory analysis confirmed that the addition of a limited amount of limonene, either in the L- or D-form, to TNG71-*Bph45* increased the attractiveness of the rice plants to BPH. Both the transcript amount of limonene synthase and the limonene content were significantly lower in TNG71-*Bph45* than in TNG71 and were not induced by BPH infestation in TNG71-*Bph45*. TN11-*Bph45* also exhibited a reduced limonene content and was less attractive to BPH, confirming the effect of the *Bph45* allele. A comparison of limonene contents in IR24 and its derived NILs revealed that the *Bph20* and *Bph18/Bph32* genes also decrease the limonene content when introgressed into IR24. This study, for the first time, demonstrates that limonene plays a critical role in the attractiveness of rice to BPH, which may be relevant to the often-observed phenomenon of rice antixenosis against BPH.

## 2. Results

### 2.1. Recurrent Parent Genome Recovery Analysis of TNG71-Bph45

A recurrent parent genome recovery analysis using 134 InDel and 173 SSR markers which polymorphic between TNG71 and *O. nivara* (IRGC 102165) was performed. Bandings differed between TNG71-*Bph45* and TNG71 were detected only on chromosome 4 (RM3636–RD04032_1), 7 (RM6767), and 11 (RD1107–RD1108) (Appendix A). TNG71-*Bph45* and TNG71 exhibited identical bandings on 292 among 307 markers analyzed. The recurrent parent genome recovery of TNG71-*Bph45* was about 95.1% and was granted as a near-isogenic line (NIL) of TNG71.

### 2.2. Host-Choice Settling Preference of BPH

To examine whether BPH has a host preference for TNG71 or TNG71-*Bph45*, host-choice tests within nylon cages were performed. In the short-distance test, BPHs were placed on Petri dishes at equal proximal distances (1–5 cm) to the surrounding pots of TNG71 or TNG71-*Bph45* rice plants. After 24 h, all BPHs on the plants were captured for counting, then placed back into the Petri dishes for retesting. Three consecutive tests were performed with three replicates. BPHs consistently showed a higher preference for TNG71 (60–65%) than TNG71-*Bph45* (35–35%) within 24 h (Figure 1A). Using the same experimental setting without disturbance for three days, BPHs continued to move from the TNG71-*Bph45* to TNG71 rice plants and finally reached a distribution of ~80% on TNG71 and ~20% on TNG71-*Bph45* (Figure 1B).

We performed a similar host-choice test but increased the distance between BPHs and plants to 20–25 cm (long-distance test). Interestingly, BPHs were equally distributed on TNG71 and TNG71-Bph45 rice plants in three consecutive tests except for the first one (Figure 1C). The preference of BPH for TNG71, which was diminished by distance, is possibly caused by volatile organic compounds (VOCs) emitted by the rice plant.

### 2.3. Y-Maze Olfactometer Examination of Settling Preference of BPH

To confirm that the settling preference of BPH was based on differences in the VOCs of rice plants but not on other characteristics such as surface structure, a Y-maze olfactometer experiment was performed in which BPHs were provided only with rice VOCs and were entrapped with no return once they made a choice. We found that the BPHs moved in the direction of TNG71 (~65%) more often than toward TNG71-*Bph45* (~35%) (Figure 2A). When both rice plants were heavily invaded by BPH previously, the BPH distributions remained the same, with ~59% moving toward TNG71 and ~41% toward TNG71-*Bph45* (Figure 2B).

### 2.4. GC-MS Characterization of the Volatile Organic Compounds

To answer the question of whether the settling preference is caused by attractive VOCs emitted by TNG71 or repellents emitted by TNG71-*Bph45*, we analyzed the compositions of rice VOCs by GC-MS. The rice plants emitted ~60 VOCs that exceeded 0.5% relative abundance, while the blank SPME probe, which was incubated in an empty glass container, showed no peaks (Figure 3A and Appendix A). An evaluation of the semi-quantitative differences between TNG71 and TNG71-*Bph45* using Student’s *t*-test revealed 13 compounds with significant differences between the two rice lines (Appendix A, blue font). One major peak with a retention time of 19.36 min was consistently more abundant in TNG71 (7.8 ± 1.4%) than in TNG71-*Bph45* (3.6 ± 1.5%) (Figure 3A,B). According to the National Institute of Standards and Technology (NIST) GC–MS library, version 2017 (https://diabloanalytical.com/products/software/nistms/nist-17-mass-spectral-library-software-components/ (accessed on 13 December 2022)), this compound shows >95% similarity with limonene.

Limonene has two chiral forms with the D-form occurring more commonly than the L-form in nature. Their odors are distinctly different: the D-form possesses an orange fragrance while the L-form has a piney, turpentine-like smell. Therefore, we characterized the forms of limonene generated by the rice plants. Using a Cyclodex-B chiral column, L-limonene was detected slightly earlier than D-limonene (Figure 4A). Co-incubation of rice plants with the authentic standards of limonene demonstrated that both TNG71 and TNG71-*Bph45* mainly emitted the L-form of limonene, as the peak overlapped with the L-form but not the D-form standard (Figure 4B,C). To estimate the amount of limonene in rice plants, a standard curve of various amounts of L-limonene versus peak area determined by GC-MS was established, and the amount of limonene emitted by 5 g of rice leaves was semi-quantified using the same GC-MS conditions with interpolation. Five grams of TNG71 and TNG71-*Bph45* rice tissues were found to emit 2.380 ± 0.555 and 0.460 ± 0.025 nL of limonene, respectively (Figure 4D); these estimates agreed with the peak intensities obtained from chiral column analysis of the SPME probes incubated with 3.125 nL of limonene standards (Figure 4A–C).

### 2.5. Confirmation of the Attractiveness of Limonene to BPH

To examine whether TNG71 attracted BPH because it contains a higher amount of limonene, we supplemented TNG71-*Bph45* with extra limonene in the Y-maze test. When switching the Y-maze settings of the two TNG71-*Bph45* rice pots, one with limonene and the other with solvent only, we consistently found that the side of the maze with added limonene attracted more BPHs (~64%) than the other side (~36%). This ratio is similar to that of TNG71 vs. TNG71-*Bph45* (Figure 2A). In terms of attractiveness to BPH, L-limonene was effective at a concentration of ~24 nL/L, but not at the lower (12 nL/L) or higher (48 nL/L) concentrations tested (Figure 5A). D-limonene was also effective, but at even lower concentrations, 1.5–3 nL/L (Figure 5B). These results are consistent with the observation that rice emits only nl-amounts of limonene per 5 g of fresh rice leaves (Figure 4D) and suggests that BPH can be attracted by both forms of limonene, with a ~10-fold lower threshold for D-limonene than for L-limonene.

It is noteworthy that limonene alone in the absence of rice plants, even at the theoretically effective concentration, did not significantly attract BPH (Figure 5C,D). Similar Y-maze experiments were also conducted for several differentially emitted VOCs (Appendix A), including the TNG71-Bph45-specific compounds methyl 2-methylbutyrate and eucalyptol and the TNG71-specific compound 3-methylthiophene. However, no significant differences in BPH distributions were found.

### 2.6. Transcript Abundance of Limonene Synthase in Rice Plants

Limonene is converted from geranyl pyrophosphate (GPP) by limonene synthase, which is located in chloroplasts. Although rice has 38 genes annotated as terpene synthase (TPS) according to rap-db (https://rapdb.dna.affrc.go.jp/) (accessed on 26 January 2021), only *OsTPS19* and *OsTPS20* have been demonstrated to be localized in chloroplasts and to be responsible for limonene production in rice [20,21]. As *OsTPS19* and *OsTPS20* share 97% nucleotide sequence identity, we examined the expression of both genes concurrently by RT-qPCR. The expression of *OsTSP19/20* was significantly higher in TNG71 than in TNG71-*Bph45* (Figure 6A, 0 h), and it was gradually induced up to 4-fold by BPH infestation in TNG71 but not in TNG71-*Bph45* (Figure 6A, 24–72 h). In addition, the amount of limonene emitted was increased by BPH infestation only in TNG71 (Figure 6B). Moreover, more limonene was emitted by plants at the flowering stage than at an earlier stage (40-day-old plant), and this trend was also observed only in TNG71 (Figure 6C). 

### 2.7. Introgression of the Bph45 Allele Affects the Limonene Content in Rice

To determine whether the low limonene content in TNG71-*Bph45* is due to the *Bph45* allele, the limonene contents in another *Bph45* introgression rice line, TN11-*Bph45*, and its parental line TN11 were compared. Indeed, TN11-*Bph45* exhibited a much lower amount of limonene (~1.9%) than TN11 (~12.3%) (Figure 7), demonstrating that the low limonene content is conferred by the *Bph45* allele.

### 2.8. The Amount of Limonene Can Be Conferred by Various BPH-Resistance Genes

As more than 44 BPH-resistance genes have been identified so far, we wondered whether they also cause a decrease of limonene content in BPH-resistant rice. We examined the limonene content in IR24 and IR24-derived NILs that harbor various BPH-resistance genes (Figure 8A) using semi-quantitative GC-MS analysis. The original IR24 line contained ~17% limonene; however, the content decreased to 6–9% in NIL-*Bph9*, NIL-*Bph17*, and NIL-*Bph18* (*p* < 0.05 or *p* < 0.01) and to 2–3% in NIL-*Bph20* and NIL-*Bph18 + 32* (*p* < 0.001). The limonene contents in the NIL-*Bph4*, -*Bph10*, -*Bph21*, -*Bph26*, -*Bph32*, -*Bph2 + 32*, and -*Bph9 + 32* lines were similar to that of IR24 (Figure 8B).

We then examined whether the host preference exhibited by BPH correlates with the limonene contents in the IR24-NIL lines. In a Y-maze olfactometer test, BPH preferred IR24 (~60%) over the NIL-*Bph20* and NIL-*Bph18 + 32* lines (~40%) (Figure 9). We conclude that decreasing the amount of limonene may be a common mechanism that can be conferred by several, although not all, BPH-resistance genes.

## 3. Discussion

### 3.1. Investigation of the Mechanism Underlying Antixenosis

Various BPH-resistance genes, which reduce the attractiveness of rice to BPH, have been found in the *indica* varieties Swarnalata, Rathu Heenati, and RP2068-18-3-5 and wild rice species such as *Oryza latifolia* and *O. rufipogon* Griff [7,14,22,23,24,25]. Nevertheless, the mechanisms underlying such antixenosis phenomenon conferred by these genes remain uninvestigated. Plant volatiles, which play important roles in the attraction or repellence of insects, may determine the insect community composition. S-linalool and (E)-β-caryophyllene are two abundant VOCs emitted by several rice varieties; S-linalool repels while (E)-β-caryophyllene attracts BPH [26]. Moreover, Lu et al. [27] reported that both 2-heptanone and 2-heptanol are emitted as repellents by BPH-infested rice. However, none of the four VOCs were emitted differentially by the two rice lines in this study (Appendix A).

### 3.2. Reduction of Limonene Emission and Attraction of BPH Simultaneously by the Bph45 Gene

We found in this study that more BPHs were distributed on the sensitive rice variety TNG71 than on the resistant line TNG71-*Bph45*, with a ratio of 6:4 at 24 h after the release of BPHs (Figure 1A). The distribution ratio gradually changed to 8:2 at about 72 h (Figure 1B). GC-MS data and the Y-maze analysis demonstrated that the lower limonene content in TNG71-*Bph45* decreases the attraction of BPHs (Figure 3 and Figure 5). Moreover, both the limonene content and *OsTPS19/20* transcript levels were induced by BPH infestation in TNG71 but not in TNG71-*Bph45* (Figure 6). We conclude that the decreased attractiveness of TNG71-*Bph45* to BPH may be due to the reduced limonene production level.

### 3.3. Attraction of BPH by Limonene at Ppb Levels Emitted by Rice

In rice, limonene is synthesized constitutively, and its emission is enhanced under abiotic and biotic stress conditions [20]. Among the various VOCs emitted by TNG71, limonene is one of the major components, constituting an average of 7.81% of the total peak area of the GC chromatogram (Appendix A). Semi-quantification using the SPME-GC-MS method revealed that 5 g TNG71 could emit about 2.38 nL limonene (Figure 4D). This amount agrees with the 1.5–24 nL/L of limonene found to attract BPHs (i.e., ppb level, Figure 5). On the other hand, our study also showed that D-limonene repelled BPHs when the concentration was above 768 nL/L (Appendix A). Although it appears that limonene plays two opposite roles, as an insect attractant and a repellent or insecticide, the difference may be attributed to the concentration applied. 

### 3.4. Attraction of BPH by Both L- and D-Limonene

In nature, limonene exists in two enantiomeric forms, D and L. Similar to other reports [20,28], we found that L-limonene was the predominant form in TNG71 and TNG71-*Bph45* (Figure 4B,C). Interestingly, D-limonene also attracted BPHs but at a ~10-fold lower concentration than L-limonene (Figure 5A,B). We suspect that both forms of limonene are present in rice and both can act as attractants to BPHs. Nevertheless, limonene alone is not sufficient to attract BPHs (Figure 5C,D), indicating that some other rice VOCs may be required.

In plants, monoterpenes (C10) are synthesized by the plastid-localized TPS. This enzyme isomerizes geranyl pyrophosphate (GPP) to (3*S*)- and (3*R*)-linalyl diphosphate (LPP) intermediates, which are in turn cyclized to L- and D-limonene, respectively. As these two LPPs are structurally distinct, this raises the question of whether their synthesis is catalyzed by different TPSs. However, a comparison of L- and D-limonene synthases isolated from plant species that produce almost pure L- and D-limonene, respectively, failed to reveal structural determinants responsible for the selection of stereochemically different substrates [29]. Moreover, Rajaonarivony et al. [30] reported that the limonene synthase from peppermint lacks strict enantiospecificity and can catalyze the conversion of both the (3*S*)- and (3*R*)-enantiomers of LPP to limonene, although the latter reaction occurs at a lower (~60%) reaction velocity. Therefore, it is possible that rice can produce both forms of limonene using the same limonene synthase.

### 3.5. Controversial Functions of Limonene, Depending on the Forms and Concentrations Applied

Limonene is a clear, colorless liquid hydrocarbon classified as a cyclic monoterpene with a wide range of uses. As one of the most common essential oil constituents of aromatic plants, limonene has been used as a non-selective organic herbicide, e.g., the product manufactured by Avenger Organics. Meanwhile, limonene is a registered active ingredient in at least 15 pesticide products [31]. The LC50 values of limonene in a 24-h fumigation test on German cockroach and rice weevil were both about 20 ppm [32]. And the LC50 was even higher, up to 300 ppm, when used on mites in the form of natural essential oil from lemon grass [33]. Opposite to roles as a repellent or even insecticide, limonene was found to act as a strong attractant to the rice herbivore *Sogatella furcifera* [34]. A similar effect was seen in *Harmonia axyridis* in which limonene at a concentration of 12.5 ppm attracted the insect but at higher concentrations repelled the insect [35].

### 3.6. Implicated Applications in Pyramid Breeding against BPH

In this study, we found that decreasing the limonene content through the introgression of *Bph45*, either into TNG71 (Figure 3B) or TN11 (Figure 7), has an antixenosis effect leading to a decrease in BPH infestation. Our results suggest that a low limonene content in rice may be used as an indicator of BPH resistance ability. Moreover, surveys of the limonene contents in BPH-resistance gene-introgressed IR24-NILs revealed that several genes including *Bph9*, *Bph17*, *Bph18*, and *Bph20* also confer rice with low limonene emission (Figure 8) coupled with low attractiveness to BPH (Figure 9), indicating at least partial similarity in their resistance mechanisms. This finding should provide valuable information for pyramid breeding, which will ideally combine BPH resistance genes that act through different mechanisms [36].

### 3.7. Implicated Applications in Integrated Pest Management against BPH

In rice, BPH infestation increases VOC emission and changes the profile of VOCs [37]. Manipulation of volatile emissions in crops has great potential for the control of pest populations. When limiting the amount of nitrogen fertilizer to meet the concept of “Green Super Rice”, which aims to reduce the cost and ecological footprint of rice cultivation, Sun et al. [38] found that a *Bph14*-containing rice line showed higher resistance to BPH-feeding concomitant with reduced emission of volatile terpene compounds including limonene. Zhang et al. [39] found that VOCs emitted by uninfested plants attract and “pull” BPH while VOCs from infested plants “push” BPH away from plants. The addition of a blend that mimics the natural composition of the infested plants reduces the attractiveness of rice to pests; therefore, it may be used for crop protection. In addition to repelling BPH, VOCs may also be used to attract artificially released parasitoids or predators of BPH [40]. As our study found that limonene attracts BPH at very low concentrations (ppb level) while repelling it at high concentrations (ppm), the manipulation of limonene content may provide an IPM strategy for rice cultivation.

### 3.8. Implicated BPH Resistance Mechanism by Bph45

In the host-choice test, although TNG71 only attracted 60–65% of the BPH population within the first 24 h (Figure 1A), BPHs continued to move onto TNG71 and reach ~80% of the population after 72 h (Figure 1B). As both rice lines did not change in attractiveness to BPH after infestation (Figure 2B), we suspect that the continued movement of BPHs from TNG71-*Bph45* to TNG71 may be attributed to other physical factors that occur after their first contact. In addition to producing less limonene, rice plants with *Bph45* may have other mechanisms conferring BPH resistance.

## 4. Materials and Methods

### 4.1. Plant Materials

TNG67, TNG71, and TN11 are *japonica* rice varieties developed and cultivated in Taiwan. TNG71 has excellent grain quality with a taro aroma [41], while TN11 is distinguished by its high yield [42]. TN11-*Bph45* was bred by a cross between TNG71-*Bph45* and TN11. Using RM3317 and RM16655 as the foreground selection markers and recurrently backcrossing with TN11, the seeds of a BC3F2 line were used in this study. A major dominant QTL, *Bph45*, originating from *O. nivara* was introgressed into TNG71 via recurrent backcrossing and marker-assisted breeding using RM3317 and RM16655 as foreground selection markers [6]. IR24 and NILs harboring various BPH-resistance genes were obtained from the International Rice Research Institute (IRRI) in Manila, the Philippines [43]. To ensure that all rice plants were grown under the same conditions for parallel comparisons, pots of plants for comparison were placed in alternating order in the same box supplied with water. Seven days after sowing, seedlings were thinned to 10 plants per 3 inch pot, and the plants were grown until they were ~40 days old.

### 4.2. DNA Preparation and Marker Analysis

Plant DNA was extracted and analyzed according to [44]. Primer information for INDEL and SSR markers can be found in [44] and the Gramene database (https://archive.gramene.org/markers/microsat/) (accessed on 12 September 2019), respectively.

### 4.3. BPH Maintenance

A stock culture of BPH biotype 1 was maintained in cages (40 × 40 × 80 cm) with potted TNG67 plants in the greenhouse facility of the Chiayi Agricultural Experiment Station of Taiwan Agricultural Research Institute (TARI), Taiwan. Adult female BPH populations were periodically transferred to cages with clean rice plants for 24 h to allow oviposition. After removal of the female BPHs, the plants with eggs were grown until the nymphs reached the 2nd–3rd instar. These nymphs were then used in this study.

### 4.4. Chemicals

Ten microliters of D-limonene (product no. 62,118, analytical standard, Sigma-Aldrich, St. Louis, MO, USA) or L-limonene (product no. 62,128, analytical standard, Sigma-Aldrich) was dissolved in 10 mL ethanol (95%) as a stock solution, which was used within one week. For the Y-maze olfactometer experiment, about 30–960 µL of stock solution was bought up to 10 mL with water, and then 2 mL of the solution was transferred onto a cotton ball beside the rice plants. The final concentration of limonene (1.5–48 nL/L) in the container was calculated by dividing the actual amount of limonene by the volume of the glass container (~4 L). Solvent blanks (cotton balls with an equal volume of solvent) served as controls. For the solid-phase microextraction-gas chromatography-mass spectrometry (SPME-GC-MS) analysis (see below), the stock solution was further diluted 10-fold to 1 nL/10 µL with ethanol (95%). Finally, 10–50 µL of the solution was used. 

### 4.5. Host-Choice Test of BPH

The host-choice test of BPH was determined using the method from Liu et al. [24] with modifications. In the short-distance test (1–5 cm between the BPH nymphs and rice), four pots of TNG71 and four pots of TNG71-*Bph45* were placed in alternation side-by-side in a 4 × 2 format. Within a BugDorm (47.5 × 47.5 × 47.5 cm^3^, BD44545, MegaView Science, Taichung, Taiwan), about 480 2nd–3rd instar nymphs were released from three Petri dishes arranged in the middle of eight rice pots. In the long-distance test (20–25 cm between the BPH nymphs and rice), two pots of TNG71 and two pots of TNG71-*Bph45* were arranged at the four corners of a BugDorm as described above. About 400 2nd–3rd instar nymphs were released from one Petri dish that was placed in the center of the BugDorm. The distribution of nymphs from the same BPH population between the two rice lines was determined every 24 h. For the short- and long-distance tests, >250 and >150 instar nymphs, respectively, were still alive and used for the calculation of the distribution in each round.

### 4.6. Y-Maze Olfactometer Examination of BPH Settling Preference

For the Y-maze olfactometer, which was assembled manually (Appendix A), a controlled flow of charcoal-cleaned air was pumped through two glass containers (~4 L each), one containing a pot of TNG71 and the other containing a pot of TNG71-*Bph45*, and then the air reached no-return traps, which were connected to the Y-tube. The three arms in the Y-tube were about 2 cm in diameter and 10 cm in length. The BPHs (~350 2nd–3rd instar nymphs) released at the bottom arm moved freely towards either arm and finally to the traps. Two days later, the BPHs that moved to the two traps were counted (usually 200–250 instar nymphs in total). The locations of the TNG71 and TNG71-*Bph45* plants were switched in repeated experiments. 

### 4.7. SPME–GC–MS Analysis of Rice Volatile Organic Compounds

Five grams of fresh rice leaf, with a minimum cutting to keep the leaves (including sheath and blade) as intact as possible, was enclosed in a 1200 mL glass container, and the VOCs were collected for 16 h at room temperature using an SPME fiber (75 μm CAR/PDMS, fused silica core, Supelco Co., Bellefonte, PA, USA). On the next day, the SPME fiber was inserted into the injection port of a GC–MS instrument (single quadrupole GCMS-QP2010 SE, Shimadzu Co., Tokyo, Japan). Hydrocarbons were separated by a capillary column of 0.25 mm i.d. × 30 m length with a 0.5-μm film (SH Rtx-5MS, Shimadzu Co., Tokyo, Japan), with helium as the carrier gas at a flow rate of 1.1 mL/min. An injection port was set at 260 °C in the split mode (ratio 5). The thermal program was set as follows: 40 °C (5 min hold), 4 °C/min up to 160 °C, then 10 °C/min until 280 °C (3 min hold). The relative concentration of each compound in rice was semi-quantified based on the peak area integrated by the analysis program. A Wiley/NBS Registry of Mass Spectral Data search and authentic reference compounds were used for substance identification. The assay was performed five times using biologically independent plants.

### 4.8. Chiral SPME–GC–MS Analysis of Rice Volatile Organic Compounds

The chirality of limonene was examined by comparing the retention times and mass spectra of rice VOCs with the authentic standards D-limonene and L-limonene using the method from Maruyama et al. [45] with modifications. A Cyclodex-B chiral capillary column of 0.25 mm i.d. × 30 m length with a 0.25-μm film (J&W, Agilent Technologies, Santa Clara, CA, USA) was used. The thermal program was set as follows: 40 °C (3 min hold), 3 °C/min up to 100 °C, then 15 °C/min until 240 °C (3 min hold). The flow rate of the carrier gas (helium) was 1 mL/min.

### 4.9. RNA Isolation and RT-qPCR

Rice plants (40-day-old) were infested with confined 2nd–3rd instar nymphs (20 nymphs/plant), then the bottom sheath (~4 cm) was collected at 0, 24, 48, 72, or 96 h post-infestation (hpi) and kept at −80 °C before use. Total RNA was extracted using the RNeasy Plant Mini Kit (Plant RNA isolation kit, GPR06, P&J Science, Taipei, Taiwan) according to the manufacturer’s instructions. RNA samples were treated with RNase-free DNase (New England Biolabs, Hitchin, UK), and PCR was performed to confirm that there were no DNA remnants. Synthesis of the first cDNA strand was performed using the PrimeScript RT Reagent Kit (TaKaRa Bio, Kusatsu, Japan).

Each real-time PCR experiment consisted of 5 μL of KAPA SYBR^®^ FAST qPCR Kit Master Mix (KAPA Biosystems, Wilmington, NC, USA), 4 μL cDNA, 0.6 μL of 10 μM primer mix, and 0.4 μL double distilled water. Each sample was analyzed in three biological replicates with two technical duplicates using the Eco^TM^ real-time PCR system (Illumina, San Diego, CA, USA) and its relative quantification program. The cycling parameters were an initial denaturation at 95 °C for 3 min, followed by 40 cycles of 95 °C for 10 s and 60 °C for 20 s. Finally, steps of 95 °C for 15 s, 55 °C for 15 s, and 95 °C for 15 s were used to generate a melting curve graph. The ubiquitin-40S ribosomal protein S27a-1-like gene of rice served as an internal quantification standard [46]. Primers for real-time PCR were designed using the Primer3Plus website (https://www.primer3plus.com/ (accessed on 13 December 2022), version: 3.2.6) and are listed in Appendix A. Double delta Ct analysis was used for evaluating gene expression. 

### 4.10. Data analysis

Independent experiments (N > 3) were performed and Student’s *t*-test in Excel (Office 2019) was used for statistical comparisons; the type 1 and type 2 parameters were used for paired and unpaired data, respectively. Statistical significance was defined as * *p* < 0.05, ** *p* < 0.01, and *** *p* < 0.001.

## 5. Conclusions

Antixenosis is a well-known phenomenon whereby a plant impedes the settling, colonization, or oviposition behaviors of insects. Nevertheless, the underlying mechanism for rice antixenosis against BPH remains unsolved. We found in this study that the *Bph45* gene simultaneously confers rice with a low limonene content and reduces its attractiveness to BPH. Several other BPH-resistance genes including *Bph9*, *17*, *18*, and *20*, are also associated with reduced limonene content, supporting the generality of the mechanism and providing an evaluation strategy that will aid decision-making in gene pyramid breeding. Moreover, as limonene attracts BPH at very low concentrations (ppb level) while repelling it at high concentrations (ppm), manipulation of limonene content may provide an Integrated Pest Management strategy for rice cultivation.

## Figures and Tables

**Figure 1 ijms-24-01798-f001:**
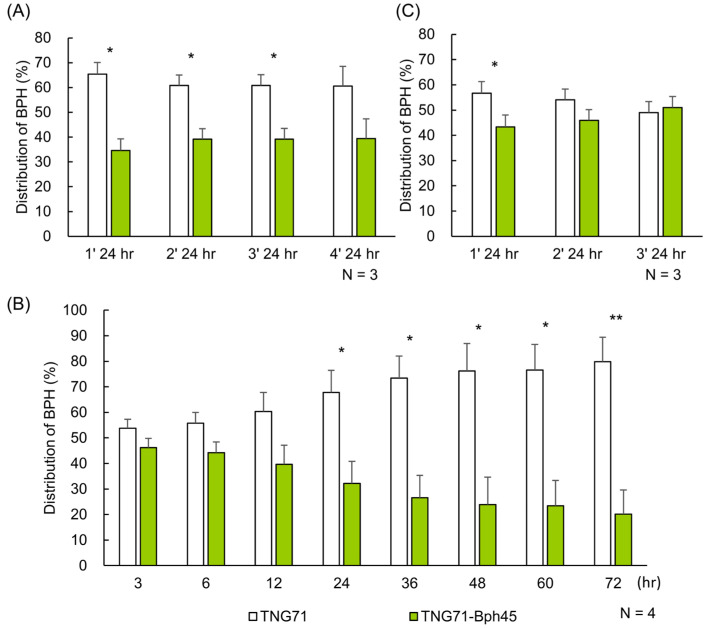
Settling preference of BPH in the host-choice test. Within a BugDorm, pots, each harboring 10 TNG71 or 10 TNG71-*Bph45* seedlings, were arranged as described in Materials and Methods. BPHs were released for 24 h and then collected to determine their distributions (**A**) Short-distance test (1–5 cm between BPH and rice). The same population of BPH was released and re-collected for a total of four rounds. (**B**) Short-distance test without disturbance. The BPHs were not collected, but their distributions between TNG71 and TNG71-*Bph45* were determined at the indicated times. (**C**) Long-distance test (20–25 cm between BPH and rice). Similar to (**A**) but the same BPH populations were tested for a total of three rounds. Statistical analysis was performed using Student’s paired *t*-test (N = 3 or N = 4). * *p* < 0.05 and ** *p* < 0.01.

**Figure 2 ijms-24-01798-f002:**
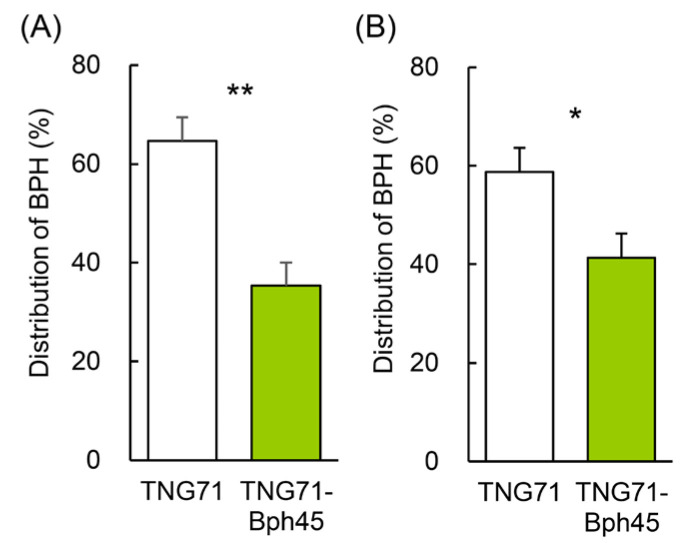
The settling preference of BPH examined using a Y-maze olfactometer. BPHs were released in a Y-maze tube with air supplied through containers containing 10 TNG71 or TNG71-*Bph45* rice plants, which were (**A**) not previously invaded and (**B**) previously invaded by 300 BPHs for 3 days. The percentage of BPHs that moved to each side of the trap was calculated 2 days after release. Statistical analysis was performed using Student’s paired *t*-test (N = 4). * *p* < 0.05 and ** *p* < 0.01.

**Figure 3 ijms-24-01798-f003:**
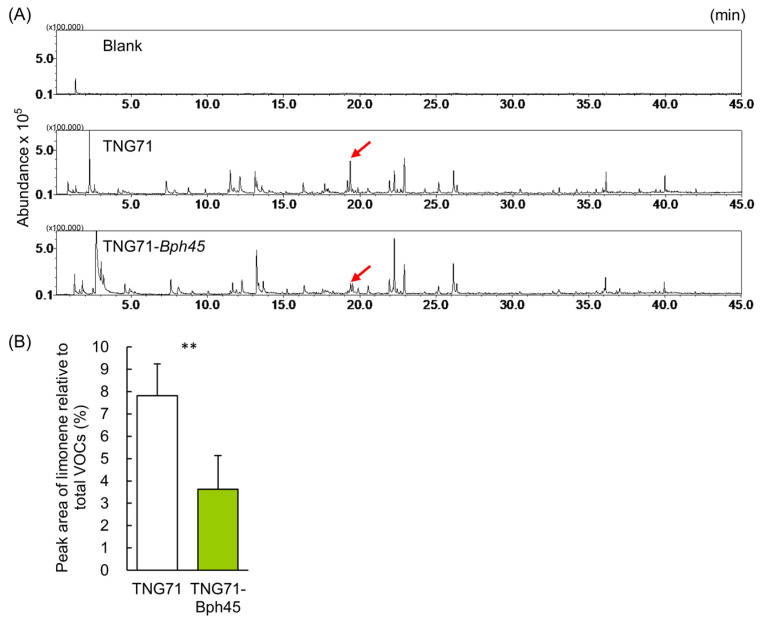
Comparison of the Volatile Organic Compounds emitted by rice lines using GC-MS. Five grams of 40-day-old rice leaves were incubated with an SPME probe overnight at room temperature and then analyzed by GC-MS. (**A**) Typical chromatograms were obtained from the blank (top), TNG71 (middle), or TNG71-*Bph45* (bottom). The red arrow indicates the peak of limonene that differed in amount consistently between TNG71 and TNG71-*Bph45*. (**B**) The relative amount of limonene was semi-quantified according to its peak area in the GC-MS profile. Statistical analysis was performed using Student’s unpaired *t*-test (N = 5). ** *p* < 0.01.

**Figure 4 ijms-24-01798-f004:**
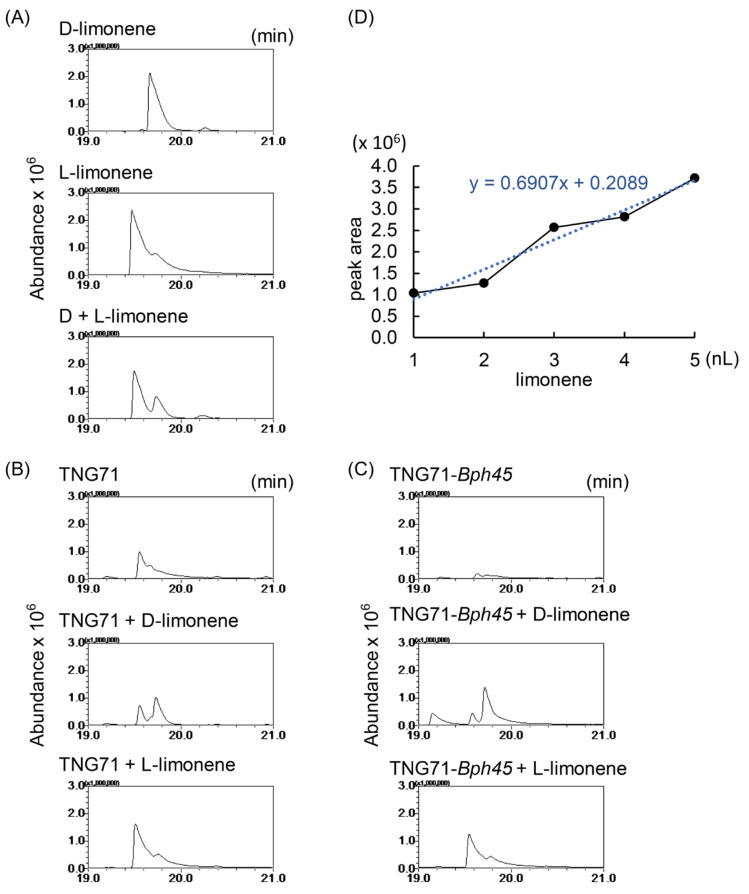
Analysis of chirality and amount of limonene emitted by rice plants. VOCs emitted from 5 g of fresh rice tissues were collected for 16 h at room temperature using an SPME probe and then analyzed by GC-MS. To characterize the form of limonene in rice, 3.125 nL of the (D)-form or (L)-form authentic limonene standard alone (**A**), together with 5 g fresh leaves of TNG71 (**B**), or together with 5 g fresh leaves of TNG71-*Bph45* (**C**), was absorbed by the SPME probe then resolved on a Cyclodex-B column. To quantify the amount of limonene in rice, various amounts of the (L)-form of limonene (1–5 nL) absorbed by the SPME probe were resolved on a Rtx-5MS column (**D**). The amount of limonene in rice was calculated by interpolation. The black dots indicate the five standard concentrations used and the blue line indicates its linear regression result.

**Figure 5 ijms-24-01798-f005:**
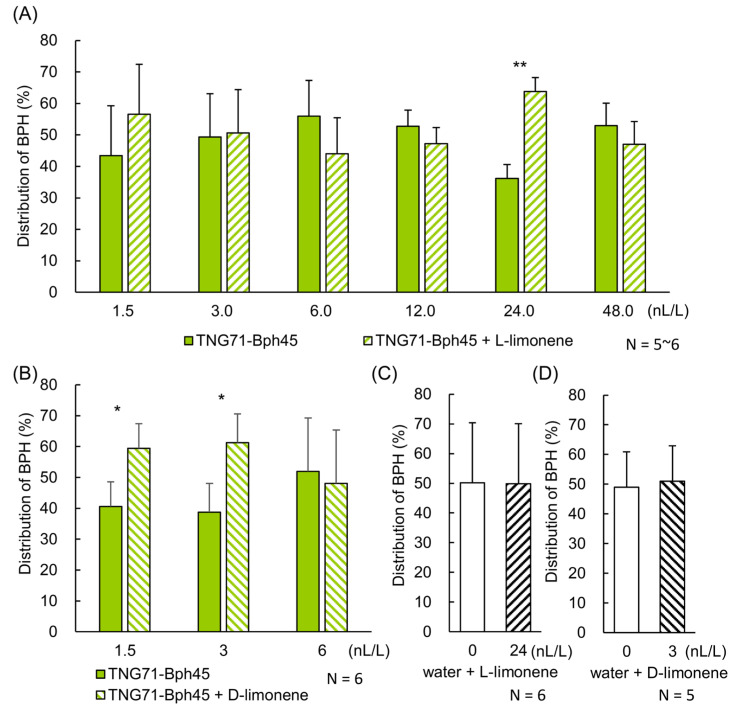
The attractiveness of limonene to BPH in Y-maze tests. BPHs were released in a Y-maze tube with air supplied through containers containing various amounts of (**A**) L-limonene or (**B**) D-limonene in the presence of TNG71-*Bph45* rice. (**C**,**D**), same as (**A**,**B**), respectively, but limonene alone was used. The percentage of BPHs that moved to each side of the trap was calculated 2 days after release. Results of independent replicates (as indicated) were analyzed using Student’s paired *t*-test. * *p* < 0.05 and ** *p* < 0.01.

**Figure 6 ijms-24-01798-f006:**
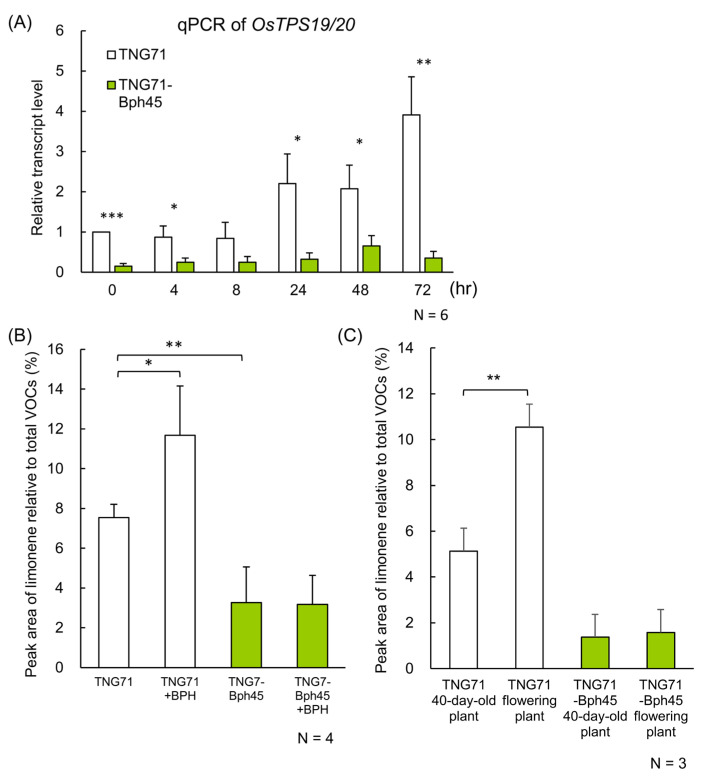
Levels of limonene synthase transcripts and limonene VOCs induced by BPH infestation and plant maturation. (**A**) The transcript amounts of TPS19/20 (which share 97% sequence identity) were quantified concurrently by RT-qPCR in rice plants after BPH infestation for various amounts of time. (**B**,**C**) As in Figure 3, the amounts of limonene semi-quantified by GC-MS were compared between rice plants without or with BPH infestation (20 instar nymphs/plant) for 72 h (**B**) or between ~40-day-old plants and flowering plants (**C**). Results of independent replicates (as indicated) were analyzed using Student’s paired *t*-test. * *p* < 0.05. ** *p* < 0.01, and *** *p* < 0.001.

**Figure 7 ijms-24-01798-f007:**
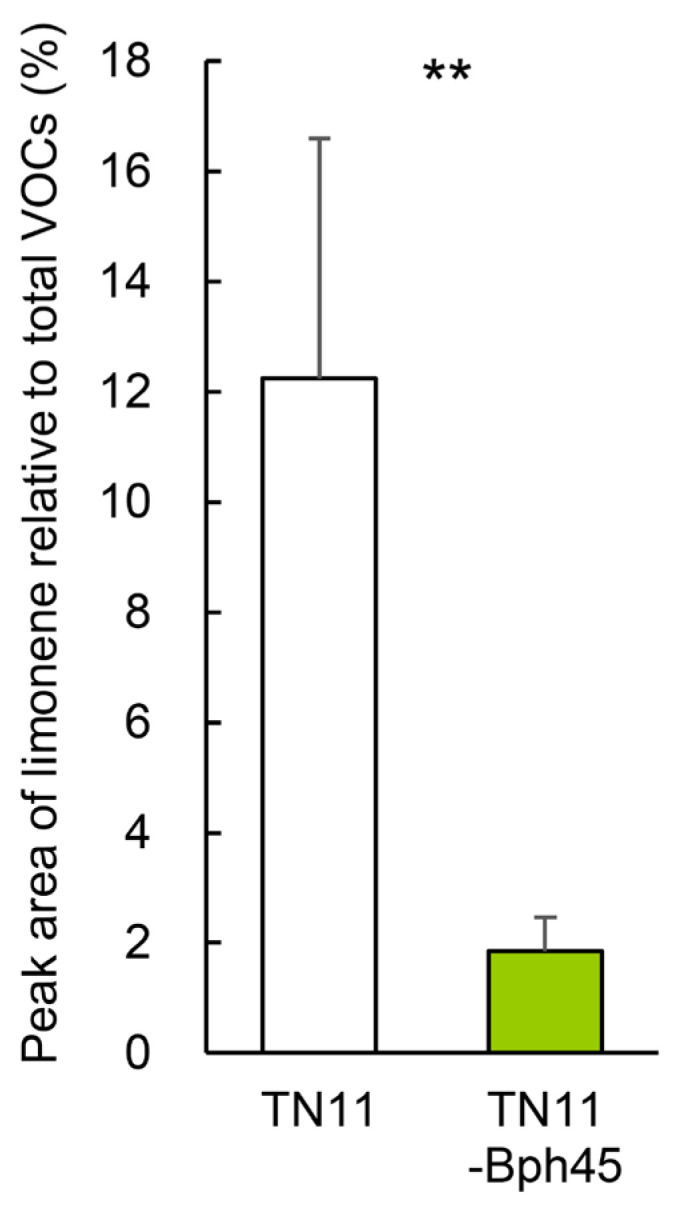
Limonene amounts in TN11 and TN11-*Bph45*. Same as in Figure 3. Statistical analysis was performed using Student’s unpaired *t*-test (N = 4). ** *p* < 0.01.

**Figure 8 ijms-24-01798-f008:**
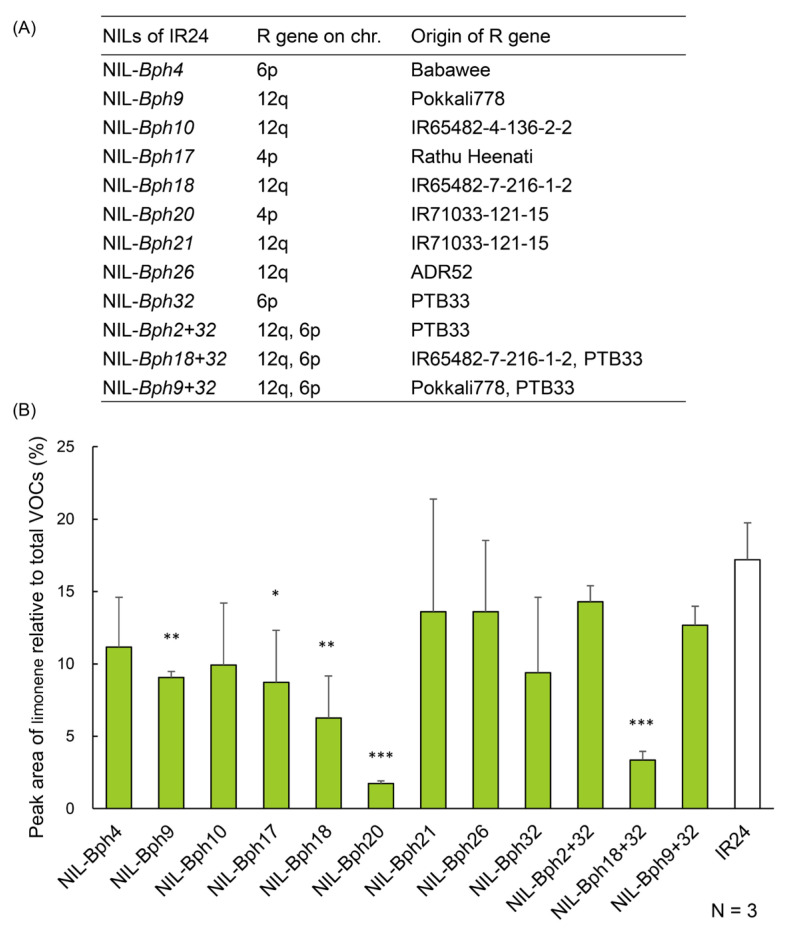
Amounts of limonene in IR24 and IR24-NILs containing various BPH-resistance genes. (**A**) IR24-derived NILs that harbor various BPH-resistance genes. (**B**) Same as in Figure 3. The amounts of limonene in the IR24 line (BPH-sensitive) and IR24-NILs (harboring various BPH-resistance genes) were semi-quantified by GC-MS. Statistical analysis between IR24 and each IR24-NIL was performed independently using Student’s unpaired *t*-test (N = 3). * *p* < 0.05, ** *p* < 0.01, and *** *p* < 0.001.

**Figure 9 ijms-24-01798-f009:**
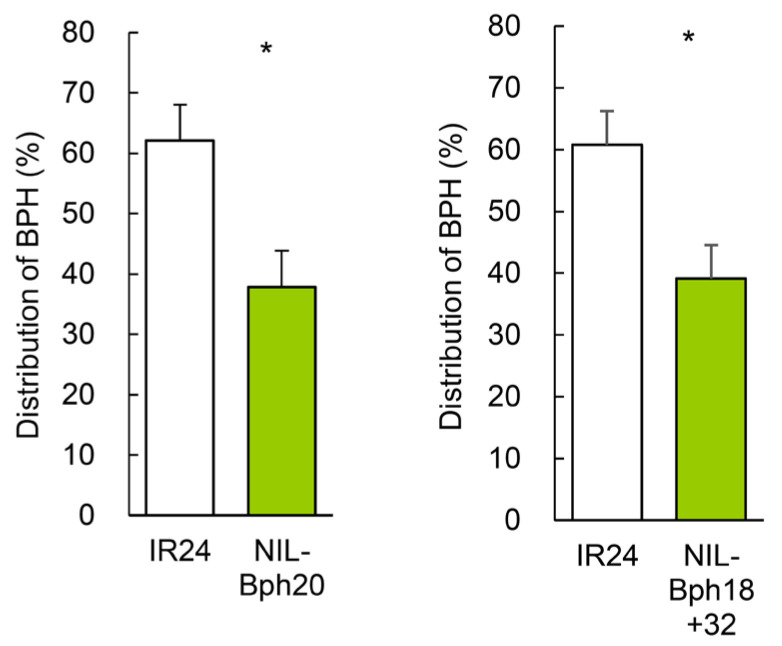
Settling preference of BPH on NIL-Bph20 or NIL-Bph18 +32 over IR24. Same as in Figure 2. Statistical analysis was performed using Student’s paired *t*-test (N = 4). * *p* < 0.05.

## Data Availability

Not applicable.

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
