# Peer review of "The Bph45 Gene Confers Resistance against Brown Planthopper in Rice by Reducing the Production of Limonene"

_ijms, 2023, doi:10.3390/ijms24021798_

Round 1

Reviewer 1 Report

The brown planthopper (BPH; Nilaparvata lugens Stål, Hemiptera, Delphacidae) is a monophagous herbivore that feeds exclusively on rice sap and causes substantial rice yield losses in Asia and even worldwide by causing direct and indirect damage. Here, Li et al compared the NIL We compared the NIL 23 (TNG71-Bph45) and the recurrent parent to explore how the Bph45 gene confers BPH resistance. They found that TNG71-Bph45 is less attractive to BPH at least partially because it produces less limonene. And exogenous application of L- and D-limonene attracted more BPH feeding on TNG71-Bph45 plants. Their study suggests that reducing limonene production may be a common resistance strategy against BPH in rice. However, several concerns should be resolved before publication in International Journal of Molecular Science.

1、          Though the recurrent parent genome recovery of TNG71-Bph45 arrived at about 95.1%, nearly 5% genome of TNG71-Bph45 are from O. nivara, which can not conclude that the difference between TNG71-Bph45 and TNG71 is due to the introgression of Bph45.

2、          The recurrent parent genome recovery of TN11-Bph45 also should been provided. Whether the results of choice test using TN11-Bph45 is similar to TNG71-Bph45? What is the expression pattern of OsTPS19/20 in TN11-Bph45 and TN11?

3、          The seedling mortality rate of TNG71-Bph45 and TN11-Bph45 should been provided. Whether exogenous application of limonene can reduce the BPH resistance level of TNG71-Bph45 and TN11-Bph45?  

Author Response

Response to Referee 1

Dear reviewer:

Thank you very much for giving us chance to revise our manuscript. We are truly grateful to you for the constructive and careful comments, which helped us very much to improve the quality of our manuscript. We have modified the manuscript accordingly. The “Track Changes” was used and the changes were marked in red in the manuscript and also listed below point by point.

Point 1、Though the recurrent parent genome recovery of TNG71-Bph45 arrived at about 95.1%, nearly 5% genome of TNG71-Bph45 are from O. nivara, which can not conclude that the difference between TNG71-Bph45 and TNG71 is due to the introgression of Bph45.

RESPONSE: We modified several sentences in the main text (lines 64-71, as shown below) to provide supporting information about the breeding rationale of TNG71-Bph45.

(lines 64-71) Genetic mapping based on a segregated population derived from O. nivara IRGC 102165 (W33) revealed a BPH resistance gene, designated Bph45 in this study. The Bph45 gene locus provided a major genetic effect on BPH resistance up to 50% Phenotypic Variation Explained (PVE), and was closely linked with SSR markers RM3317 and RM16655 on chromosome 4 [6]. One of the BPH resistance introgression lines ‘852T034’ carrying the Bph45 gene was used as the donor parent to cross with a susceptible Taiwan elite japonica variety ‘Tainung 71’ (TNG71) to establish its near-isogenic line 'TNG71-Bph45'.

Moreover, by two pieces of molecular evidence presented below, we narrowed the differential chromosomal regions between TNG71-Bph45 and TNG71 from 5% down to 2.75%.

(1) There are two main fragments originating from O. nivara in TNG71-Bph45, one on chr. 4 and the other on chr. 11 as shown by Figure S1. To exclude the involvement of chr. 11 on the limonene content, we did a screen to obtain the segregated rice progenies from the experimental field which owe homozygous alleles on chr.4/chr11, noted as S/S, N/N, N/S, and S/N (S: sativa; N: nivara). We found that only the N/N and N/S rice exhibited a low limonene content (1~2%) while both S/S and S/N rice exhibited a high limonene content (>9%). Therefore, genes on chr.4 but not chr.11 play roles in the control of limonene content.

(2) To define the O. nivara-substituted region on chr. 4 of TNG71-Bph45, we did a genomic scan along chr. 4 using INDEL markers. We found a 10.29 Mb region, delimited between Indel419 (6.25 Mb) and Indel981 (16.54 Mb), was from O. nivara. As the haploid genome size of rice is 373.2 Mb, this region accounts for 2.75% genome in TNG71-Bph45.

Point 2-1、The recurrent parent genome recovery of TN11-Bph45 also should been provided.

RESPONSE: We added several sentences in the materials and methods (into lines 379, as shown below) to provide clearer information regarding how Bph45 was introgressed into TN11. However, since both TNG71-Bph45 and TN11 are japonica rice, quite similar in their genetic background. It is hard to find polymorphic markers to perform the background selection during breeding and calculate the genome recovery rate by whole genome scan. Therefore we did not accomplish this characterization. Instead of claiming the NIL status of TN11-Bph45, we use the “Bph45 introgression rice line” to describe it.

(Inserted to line 379) TN11-Bph45 was bred by a cross between TNG71-Bph45 and TN11. Using RM3317 and RM16655 as the foreground selection markers and recurrently backcrossing with TN11, the seeds of a BC3F2 line were used in this study.

Point 2-2、Whether the results of choice test using TN11-Bph45 is similar to TNG71-Bph45?

RESPONSE: No we did not perform this experiment yet. As the throughput of Y-tube experiment is very limited, considering that both TN11 and TNG71 belong to the japonica group and are likely, exhibiting a similar phenotype in the Bph45-introgressed gene, we choose to examine the IR24 and its NILs instead at that time. However, we will keep your suggestions in mind. We believed it will be a good topic in our future work

Point 2-3、What is the expression pattern of OsTPS19/20 in TN11-Bph45 and TN11?

RESPONSE: We did not examine this point yet. Introgression of the Bph45 gene into the TN11-Bph45 line may reduce the expression of OsTPS19/20 and finally affect limonene production. We confirmed the reduction of the end-product already, and it should be worth investigating the upstream players, OsTPS19/20, and will be incorporated into our future study.

Point 3-1、The seedling mortality rate of TNG71-Bph45 and TN11-Bph45 should been provided.

RESPONSE: We did not perform this experiment but in general, the seed germination and survival rate are pretty similar between TNG71-Bph45 and TNG71. Limonene, as a native VOC that attractant BPH in rice, is involved in the antixenosis phenomenon in rice which results in a decrease in BPH infestation. As the limonene concentration is within nl/g of rice, there is no expectation to affect the survival rate of rice seedlings.

Point 3-2、Whether exogenous application of limonene can reduce the BPH resistance level of TNG71-Bph45 and TN11-Bph45?

RESPONSE: According to our results, the exogenous application of limonene to TNG71-Bph45 or TNG11-Bph45 may attract more BPH, if within a narrow range of concentration (Figure 5), to feed on the rice plant and consequently reduce its BPH resistance.

Sincerely,

Dr. Wei-Ming Leu

Associate Professor, Graduate Institute of Biotechnology

National Chung Hsing University

Phone: 886-4-22840328 ext. 767

Fax: 886-4-22853527

E-mail: wmleu@nchu.edu.tw

Reviewer 2 Report

Dear Authors, 

The present study titled “The Bph45 Gene Confers Resistance Against Brown Planthopper in Rice by Reducing the Production of Limonene” is presented in detailed well-elaborated research.

The literature review is quite good and is founded in the existing subject literature. Generally, I believe that the Author(s) provided solid theoretical foundations for the analysis using appropriate references.

But the article has a lot of content to revise, my comments related to improving the article are as follows:

Point 1: The authors hold that Bph45 enhances rice resistance to brown planthopper by reducing the production of limonene, but in line 212-213, which limonene alone in the absence of rice plants, even at the theoretical effective concentration, did not significantly attract BPH (Figure 5C and 5D), how to explain these paradoxical results?

Point 2: Whether the resistance of the near-isogenic line 'TNG71-Bph45' is due to its own cause other than the content of Limonene, such as the thickness of the cell wall, as so on. Are there any pictures of paraffin sections of the stem?

Point 3: In line 161-162, the authors revealed 13 compounds with significant differences between the two rice lines (Table S1, blue font), All 13 compounds should be tested to determine whether the content of Limonene is responsible for the change in resistance of the near-isogenic line 'TNG71-Bph45'.

Point 4: There are too many styles of bar chart. It is suggested to change it to be consistent.

Point 5: All of the tables in your paper should be changed to the standard three-line table.

Point 6: The reference format is inconsistent, such as reference 29, 31, 39, 43-46, and so on. Please check on your reference format carefully and modify it.

Round 2

Reviewer 2 Report

The authors have addressed all my comments and accordingly revised the manuscript. I have no further comment on the current version of manuscript.